# Broadband transparent optical phase change materials for high-performance nonvolatile photonics

Yifei Zhang [1,8], Jeffrey B. Chou[2,8], Junying Li [3,8], Huashan Li [4,8], Qingyang Du[1], Anupama Yadav [5], Si Zhou[6], Mikhail Y. Shalaginov[1], Zhuoran Fang[1], Huikai Zhong[1], Christopher Roberts [2], Paul Robinson[2], Bridget Bohlin[2], Carlos Ríos [1], Hongtao Lin [7], Myungkoo Kang[5], Tian Gu[1], Jamie Warner[6], Vladimir Liberman[2], Kathleen Richardson[5] & Juejun Hu[1]

Optical phase change materials (O-PCMs), a unique group of materials featuring exceptional optical property contrast upon a solid-state phase transition, have found widespread adoption in photonic applications such as switches, routers and reconfigurable meta-optics. Current O-PCMs, such as Ge–Sb–Te (GST), exhibit large contrast of both refractive index ($\Delta n$) and optical loss ($\Delta k$), simultaneously. The coupling of both optical properties fundamentally limits the performance of many applications. Here we introduce a new class of O-PCMs based on Ge–Sb–Se–Te (GSST) which breaks this traditional coupling. The optimized alloy, $Ge_2Sb_2Se_4Te_1$, combines broadband transparency (1–18.5 μm), large optical contrast ($\Delta n = 2.0$), and significantly improved glass forming ability, enabling an entirely new range of infrared and thermal photonic devices. We further demonstrate nonvolatile integrated optical switches with record low loss and large contrast ratio and an electrically-addressed spatial light modulator pixel, thereby validating its promise as a material for scalable nonvolatile photonics.

[1] Department of Materials Science & Engineering, Massachusetts Institute of Technology, Cambridge, MA, USA. [2] Lincoln Laboratory, Massachusetts Institute of Technology, Lexington, MA, USA. [3] Shanghai Key Laboratory of Modern Optical Systems, College of Optical-Electrical and Computer Engineering, University of Shanghai for Science and Technology, Shanghai, China. [4] School of Physics, Sun Yat-sen University, Guangzhou, China. [5] The College of Optics & Photonics, Department of Materials Science and Engineering, University of Central Florida, Orlando, FL, USA. [6] Department of Materials, University of Oxford, Oxford, UK. [7] College of Information Science & Electronic Engineering, Zhejiang University, Hangzhou, China. [8]These authors contributed equally: Yifei Zhang, Jeffrey B. Chou, Junying Li, Huashan Li. Correspondence and requests for materials should be addressed to J.B.C. (email: jeff.chou@ll.mit.edu) or to J.H. (email: hujuejun@mit.edu)

When optical phase change materials (O-PCMs) undergo solid-state phase transition, their optical properties are significantly altered. This singular behavior, identified in a handful of chalcogenide alloys exemplified by the Ge–Sb–Te (GST) family[1], has been exploited in a wide range of photonic devices including optical switches[2–9], non-volatile display[10], reconfigurable meta-optics[11–17], tunable emitters and absorbers[18–20], photonic memories[21–24], and all-optical computers[25]. To date, these devices only leverage phase change materials originally developed for electronic switching. Optical property modulation in these classical phase change material systems stems from a change in bonding configuration[26–28] accompanied by a metal-insulator transition (MIT)[29]. The introduction of large amounts of free carriers in the metallic or conductive state, while essential to conferring conductivity contrast for electronic applications, results in excessive loss increase due to free carrier absorption (FCA). The concurrent index and loss changes fundamentally limit the scope of many optical applications. Breaking such coupling allows independent control of the phase and amplitude of light waves, a "Holy Grail" for optical engineers that opens up numerous applications including ultra-compact and low-loss modulators[30], tunable thermal emission[31] and radiative cooling[32], beam steering using phase-only modulation[33], and large-scale photonic deep neural network[34]. The decoupling of the two effects is customarily characterized using the material figure-of-merit (FOM), expressed as:

$$\text{FOM} = \frac{\Delta n}{\Delta k}, \quad (1)$$

where $\Delta n$ and $\Delta k$ denote the real and imaginary parts of refractive index change induced by the phase transition, respectively. It has been shown that this generic FOM quantitatively correlates with the performance of many different classes of photonic devices[35–38]. Current O-PCMs suffer from poor FOM's on the order of unity, imposing a major hurdle towards their deployment in these applications.

Besides their low FOM, the limited size of the switching volume poses an additional challenge for existing chalcogenide O-PCMs. The poor amorphous phase stability of GST mandates a high cooling rate in the order of $10^{10}$ °C/s to ensure complete re-amorphization during melt quenching[39], which coupled with their low thermal conductivity[40] stipulates a film thickness of around 100 nm or less. This geometric constraint is required if complete, reversible switching is to be achieved. While not an issue for today's deeply scaled electronic memories, it constrains optical devices to ultra-thin film designs.

In this article, we report experimental demonstration of the vast capabilities enabled by an O-PCM Ge–Sb–Se–Te (GSST). GSST possesses an unprecedented broadband optical transparency and exceptionally large FOM throughout almost the entire infrared spectrum. The material therefore represents a new class of O-PCMs where the phase transition only triggers refractive index modulation without the loss penalty. It is anticipated that isoelectronic substitution of Te by Se tends to increase the optical bandgap and thus serves to mitigate the interband absorption in the near-infrared. The impact of Se substitution on FCA, which dictates the optical loss in the mid-infrared, is a main topic of investigation in this paper. We also note that while Se doping in phase change alloys has been previously investigated[41–45], their singular optical behavior has not been explored or investigated. Our work reveals that the remarkable low-loss performance benefits from blue-shifted interband transitions as well as minimal FCA, substantiated through coupled first-principle modeling and experimental characterization. Record low losses in non-volatile photonic circuits and electrical pixelated switching are

demonstrated capitalizing on the extraordinary optical properties of this new O-PCM.

## Results

**Density functional theory (DFT) modeling.** We use DFT computations to predict the phase and electronic structures of alloys in the GSST family and reveal promising trends arising from Se substitution. We have investigated the $Ge_2Sb_2Se_xTe_{5-x}$ ($x = 0$ to 5) system, the Se-substituted counterparts of the archetypal phase change alloy $Ge_2Sb_2Te_5$ (GST-225). Substitution of Te by the lighter Se atoms is believed to lead to increased bandgap and hence lessened loss in the near-infrared. However, the loss decrease has to be traded off with undesirable traits such as reduced optical contrast. The objective of the DFT model, therefore, is to elucidate the impact of Se substitution on the structural, electronic and optical properties of the $Ge_2Sb_2Se_xTe_{5-x}$ family for O-PCM applications.

We start by constructing atomic models of the $Ge_2Sb_2Se_xTe_{5-x}$ alloys (Fig. 1a–c) following procedures detailed in Supplementary Note 1, and investigate the basic phase transition behavior of GSST alloys. As shown in Fig. 1d, the cohesive energy difference between the hexagonal and cubic phases is barely affected by Se substitution for Te, hinting a cubic to hexagonal transition path in GSST resembling that of GST-225. The alloy $Ge_2Sb_2Se_5$, on the other hand, exhibits a distinctive orthorhombic structure which is stabilized by the formation of strong Se–Ge/Sb bonds (Supplementary Note 1).

To evaluate the impact of Se substitution on optical properties, we further simulate the electronic band structure of the composition group. Electronic structures modeled by DFT (Fig. 2a) confirm the semiconductor nature of all alloys. The electronic structure is preserved except in $Ge_2Sb_2Se_5$ (Fig. 2b, c, Supplementary Note 2). The bandgap increases with Se addition from 0.1 eV for GST-225 to 0.3 eV for GSS4T1, suggesting a reduction of optical loss in the near-infrared. The density of states (DOS) peaks is also weakened with increasing Se concentration, contributing to additional absorption suppression. For the orthorhombic $Ge_2Sb_2Se_5$, the theory predicts very weak absorption given its larger bandgap (0.6 eV) than that of the hexagonal phase (0.3 eV). This is expected from the lacking of half-filled degenerate orbitals due to $p$-orbital misalignment[26]. The charge density distributions (Fig. 2f, g) further reveal that misalignment between $p$-orbitals due to the surface curvature of atomic blocks and large amount of interstitial sites in the orthorhombic $Ge_2Sb_2Se_5$ phase eliminates the resonance bonding mechanism. As resonant bonding has been associated with the large optical contrast of O-PCMs[46,47], significantly diminished optical contrast is inferred for $Ge_2Sb_2Se_5$.

In summary, the DFT model suggests $Ge_2Sb_2Se_4Te_1$ (GSS4T1) as the preferred O-PCM among the compositions investigated. GSS4T1 inherits the resonant bonding mechanism essential for large $\Delta n$, and also benefits from reduced interband optical absorption with lessened near-infrared loss. On the other hand, despite its low optical loss, the structurally distinct $Ge_2Sb_2Se_5$ is limited by its diminished optical contrast. These theoretical insights are validated by experimental results elaborated in the following section. It is noted that the DFT model does not account for free carrier effects and are complemented by critical experimental studies of the materials' carrier transport and optical properties. These findings are detailed below.

**Structural, electronic, and optical properties of GSST alloys.** In order to experimentally confirm and understand the crystal structure of the various GSST compositions, a series of $Ge_2Sb_2Se_xTe_{5-x}$ ($x = 0–5$) films were prepared. Supplementary

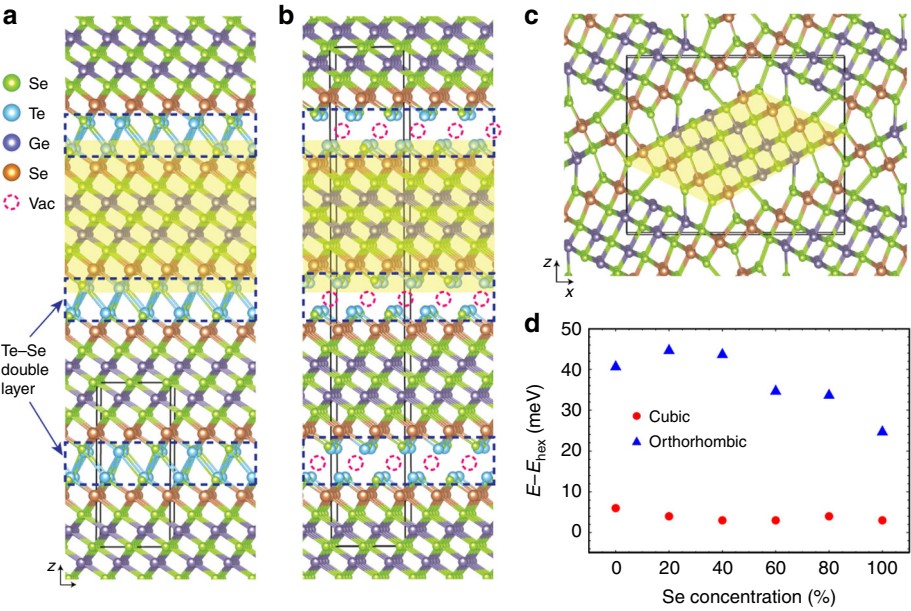

**Fig. 1** Impact of Se substitution revealed by density functional theory (DFT) simulations. Atomic structures of **a** hexagonal $Ge_2Sb_2Se_4Te_1$ (GSS4T1); **b** cubic GSS4T1; and **c** orthorhombic $Ge_2Sb_2Se_5$ with the representative atomic blocks highlighted by the yellow shaded areas. Unit cells, loosely bound Te/Se double layers, and aggregated vacancies are presented by the black boxes, dashed rectangle, and dashed circles, respectively. **d** Cohesive energies of the cubic and orthorhombic phases relative to their hexagonal counterparts with various Se concentrations

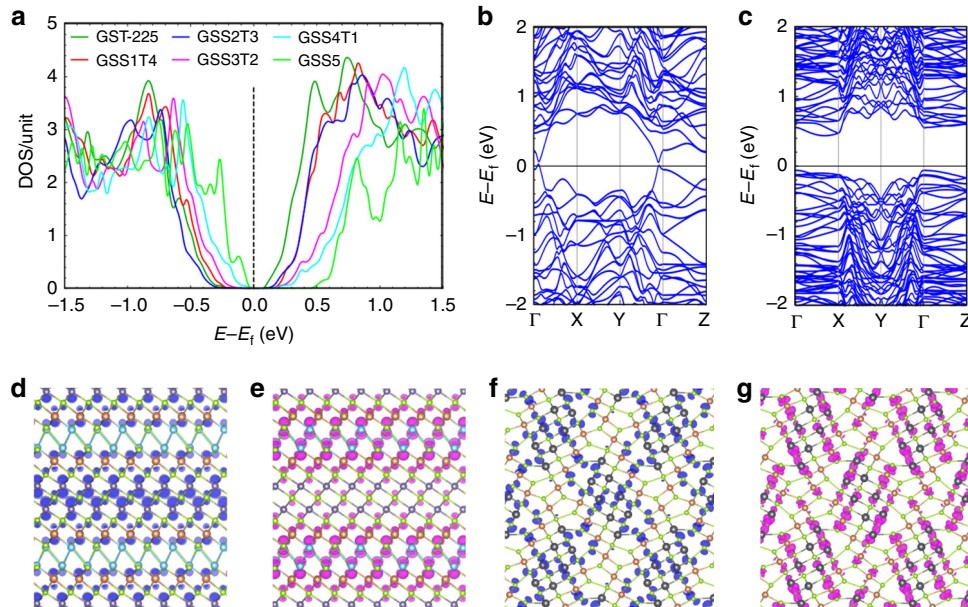

**Fig. 2** Comparison of electronic structures of hexagonal and orthorhombic phases. **a** DOS of hexagonal Ge–Sb–Se–Te and orthorhombic $Ge_2Sb_2Se_5$, with the Fermi level illustrated by the dashed line. Band structures of **b** hexagonal $Ge_2Sb_2Se_4Te_1$ (GSS4T1); and **c** orthorhombic $Ge_2Sb_2Se_5$. Charge densities of **d**, **f** valence band maximum (VBM) in blue and **e**, **g** conduction band minimum (CBM) in magenta of **d**, **e** hexagonal GSS4T1 and **f**, **g** orthorhombic $Ge_2Sb_2Se_5$

Fig. 7 in Supplementary Note 3 present X-ray diffraction (XRD) spectra of the films annealed at different temperatures. All as-deposited films are amorphous. For films with $x = 0$–$4$, annealing induces a nucleation-dominated phase change where the films first crystallize into a metastable phase followed by complete transition to the stable hexagonal structure. The crystallization onset temperature progressively increases with Se substitution, signaling enhanced amorphous phase stability. The intermediate temperature regime for the metastable phase also diminishes with

increasing Se substitution. On the other hand, $Ge_2Sb_2Se_5$ undergoes a sluggish growth-dominated transformation into an orthorhombic equilibrium phase confirmed with selected area electron diffraction (SAED) measurement (Supplementary Note 4). These findings are in excellent agreement with our theoretical predictions. The DFT modeled structures are further corroborated by quantitative fitting of the XRD spectra. For instance, DFT predicts lattice constants of $a = 4.04$ Å and $c = 16.08$ Å for hexagonal GSS4T1 whereas XRD fitting yields

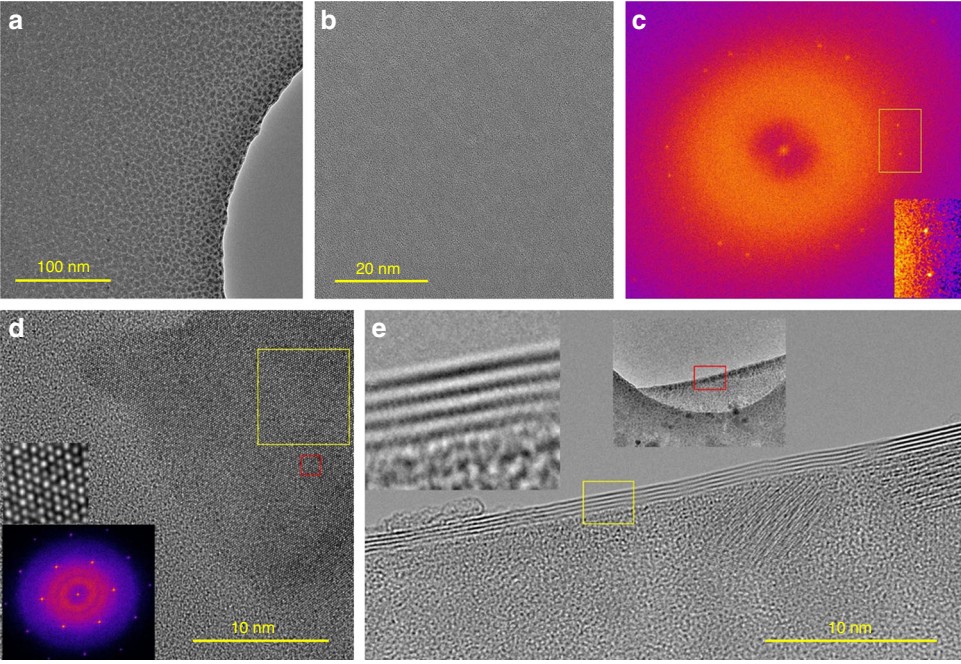

**Fig. 3** In-situ TEM analysis of the crystallization process of Ge$_2$Sb$_2$Se$_4$Te$_1$ (GSS4T1). **a** Low-magnification and **b** higher-magnification images of GSS4T1 film on a SiN holder after heating at 400 °C for 5 min. **c** Local fast Fourier transformation (FFT) of **b** showing two sets of reciprocal lattice points, which reveals that the sample contains two sets of hexagonal reflexes with a twist angle. **d** High-magnification image of the film after further annealing at 500 °C for 10 min. FFT analysis of the yellow square region shown in the inset indicates absence of the rotational stacking fault observed in **b**. **e** A back-folded region of the film suspending over a hole in the SiN support (corresponding to the red rectangle in the inset), where the layered structure of hexagonal GSS4T1 is evident

$a = 4.08$ Å and $c = 16.08$ Å. Such agreement is excellent considering that tensile strains in the order of 1% have been measured in thermally crystallized O-PCM films[48,49].

The phase transition process of GSS4T1 was further investigated in situ using aberration-corrected TEM. The as-deposited film was amorphous without visible lattice structure. Figure 3a shows a low-magnification image of GSS4T1 film on a silicon nitride (SiN) holder after heating at a nominal temperature of 400 °C for 5 min. Granular contrast is observed, and the higher-magnification TEM image (Fig. 3b) shows lattice structure detectable by a fast Fourier transform (FFT) analysis. The FFT in Fig. 3c reveals that the sample has two sets of hexagonal reflexes with a twist angle between them, indicating a rotation stacking fault between two crystals. We computed the local FFT images around different regions in Fig. 3b, and all showed the same pattern suggesting that the rotational twist occurs in the out-of-plane $z$-direction and not as lateral grains. This finding suggests that GSS4T1 forms a layered compound with an initial orientation mismatch between vertically stacked layers.

The temperature was subsequently raised to 500 °C for 10 min before cooling down to stop further transformation during TEM examination. Figure 3d shows a high-magnification image of a well-crystallized area with strong lattice contrast visible. Similar single-crystal patterns were observed across the entire sample in FFT, signifying that the rotational misorientation present at 400 °C has been removed by high temperature annealing. We also located a region where the film was suspended over a hole in the SiN support and had folded on itself (Fig. 3e). The back-folded region shows multiple lines of contrast in its profile similar to back-folded layered 2D materials[50], affirming the layered structure of hexagonal GSS4T1.

The observed layered structure is consonant with the DFT model depicted in Fig. 1a. The rotational stacking fault most likely occurs at the Se–Te double layer where the bonding is weak and many high-symmetry interfacial configurations exist as local energy minima (Supplementary Note 5). At elevated temperatures, thermal fluctuation enables the system to explore a large range of configurational space, and eventually drives it towards the global minimum, i.e., a single-crystal-like structure.

In order to experimentally verify the reduction in FCA with Se substitution, both electronic and optical measurements were performed. Electronic transport properties of the GSST alloys were studied using Hall measurement. In situ conductivity measurement during annealing (Fig. 4a) indicates that the electrical resistivity of GSST sharply drops coinciding with occurrence of phase transitions, followed by continuous decrease as annealing temperature rises due to vacancy ordering[51]. Room-temperature resistivity of GSS4T1 is over two orders of magnitude larger compared to that of GST (Fig. 4b). For all compositions, the crystalline materials show p-type conduction similar to that of the prototypical GST-225 alloy[52] with relatively minor change in Hall carrier concentrations (Fig. 4c). The drastic resistivity increase with Se substitution is therefore mostly attributed to the reduction of carrier mobility (Fig. 4d). This is possibly a consequence of the negligible energy penalties associated with structural perturbations within the Te/Se double layer predicted by our DFT simulations (Supplementary Fig. 3), which results in pronounced atomic disorder and decoupling of directional $p$-bonds. Unlike GST-225, c-GSS4T1 consistently exhibits negative temperature coefficients of resistivity (TCR) for all annealing temperatures (Fig. 4e), signaling non-metallic behavior of c-GSS4T1 (Supplementary Note 6)[29].

Although such elevated resistivity is critical to suppressing FCA, the fact that c-GSS4T1 behaves as an insulator with a negative TCR raises the question of whether large optical property contrast, the hallmark of O-PCMs, can be maintained in the absence of an MIT. To address this question, the Kramers–Kronig consistent optical constants of GSST alloys

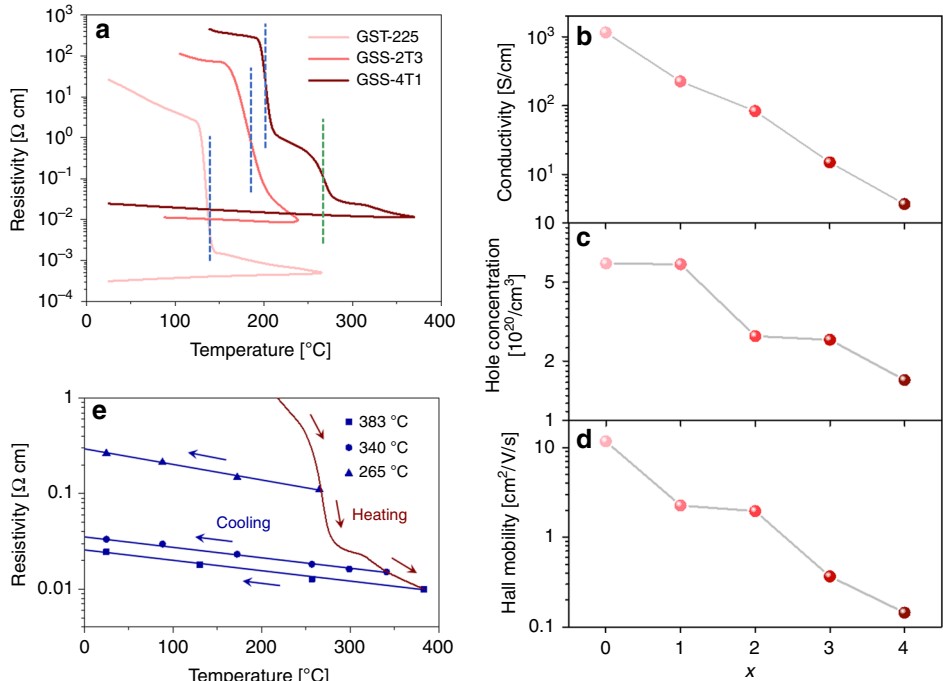

**Fig. 4** Electronic properties of $Ge_2Sb_2Se_xTe_{5-x}$ alloys. **a** Temperature dependence of resistivity of $Ge_2Sb_2Te_5$, $Ge_2Sb_2Se_2Te_3$, and $Ge_2Sb_2Se_4Te_1$ (GSS4T1) upon annealing: the distinct drop marked by blue dotted lines correspond to crystallization of the amorphous phase to the metastable cubic phase, whereas the green dotted line labels the transition towards the stable hexagonal phase. **b** Hall conductivity, **c** hole concentration, and **d** Hall mobility of $c$-$Ge_2Sb_2Se_xTe_{5-x}$; for all compositions, the films were annealed 50 °C above the amorphous-to-cubic transition temperature. **e** Temperature-dependent resistivity of GSS4T1 annealed at different three temperatures: 265, 340, and 383 °C. The temperature coefficients of resistivity are negative in all cases evidenced by the negative slope of the cooling curves

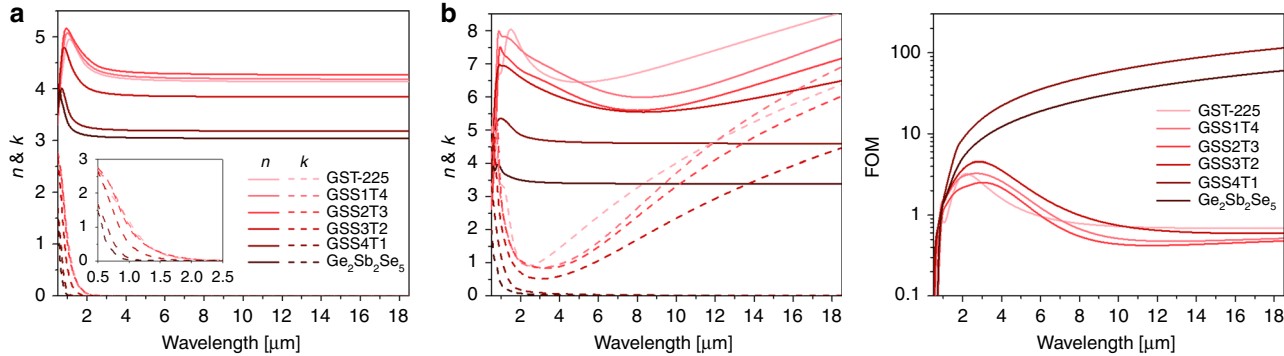

**Fig. 5** Optical properties of $Ge_2Sb_2Se_xTe_{5-x}$ films. **a**, **b** Measured real ($n$) and imaginary ($k$) parts of refractive indices of the **a** amorphous and **b** crystalline alloys. **c** Material FOM's

were obtained using coupled spectroscopic ellipsometry and transmittance/reflectance measurements from the visible through long-wave infrared (Fig. 5a, b). GSS4T1 exhibits a large $\Delta n$ of 2.1 to 1.7 across the near- to mid-infrared bands, suggesting that MIT is not a prerequisite for O-PCMs. Moreover, its remarkably broad transparency window (1–18.5 μm) owing to blue-shifted band edge and minimal FCA yields a FOM two orders of magnitude larger than those of GST and other GSST compositions (Fig. 5c). Although $Ge_2Sb_2Se_5$ similarly exhibits broadband optical transparency, its FOM is inferior to that of GSS4T1 due to its low index contrast.

**High-performance non-volatile integrated photonic switch demonstration.** Integrated optical switches are essential

components of photonic integrated circuits. Traditional optical switches mostly operate on miniscule electro-optic or thermo-optic effects (typical $\Delta n < 0.01$), thereby demanding long device lengths. The large index contrast furnished by O-PCMs potentially allows drastic size down scaling of these devices. However, at the 1550 nm telecommunication wavelength, the traditional GST-225 exhibit a low FOM of 2.0. Consequently, optical switches based on GST-225 only provide moderate contrast ratio (CR) and undesirable insertion losses (IL)[2–8]. Compounded parasitic losses and crosstalk preclude scalable integration of these individual devices to form large-scale, functional photonic circuits. Here we exploit the exceptional FOM of GSS4T1 to realize a non-volatile photonic switch with unprecedented high performance. Figure 6a shows an image of the switch device comprising a SiN racetrack resonator coupled to a bus waveguide. A 50-nm

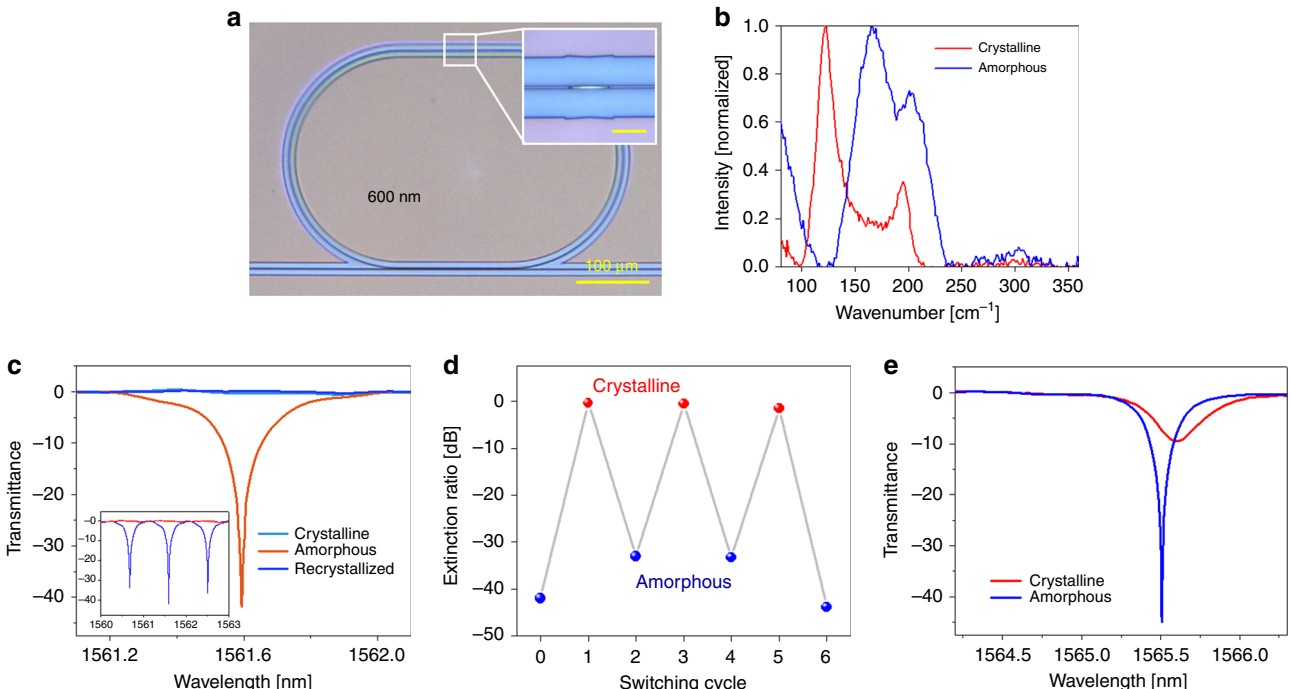

**Fig. 6** Non-volatile integrated photonic switches based on optical phase change materials (O-PCMs). **a** Optical micrograph of the resonant switch: inset shows the $Ge_2Sb_2Se_4Te_1$ (GSS4T1) strip on top of the SiN waveguide. **b** Raman spectra of laser switched GSS4T1, where the peaks at 160 and 120 $cm^{-1}$ are signatures of the amorphous and crystalline states, respectively. **c** Normalized transmittance spectra of the resonant switch integrated with GSS4T1, showing complete on/off modulation of the resonant peaks. Inset displays the broadband transmittance spectra of the same device. The three spectra correspond to three states of GSS4T1: (orange) as-deposited amorphous, (light blue) crystallized through furnace annealing, and (blue) laser recrystallized. **d** Resonance extinction ratio modulation of the device upon laser switching. **e** Normalized transmittance spectra of a reference switch device integrated with $Ge_2Sb_2Te_5$

thick strip of GSS4T1 was deposited on the resonator as illustrated in the inset. Phase transition of GSS4T1 was actuated using normal-incident laser pulses and confirmed via Raman spectroscopy. Figure 6b plot the Raman spectra of the GSS4T1 strip in both structural states, where the peaks at 160 and 120 $cm^{-1}$ are signatures of the amorphous and crystalline states, respectively. The optical property change turns on/off resonant transmission through the switch reversibly over multiple cycles, evidenced by the measured transmittance spectra in Fig. 6c and the corresponding extinction ratio modulation (Fig. 6d). The device exhibits a large switching CR of 42 dB and a low IL of <0.5 dB, outperforming all previous non-volatile switches[2–8] as well as devices based on the traditional GST-225 material with a similar configuration (Fig. 6e); as can be seen from Fig. 6d, the resonance peak is not completely turned off even when GST-225 is transformed into the crystalline state. Such remarkable performance is consistent with theoretical predictions based on the measured optical constants of GSS4T1 (Supplementary Note 7) and is attributed to its exceptional FOM.

**Pixel-level electrothermal switch for free-space reflection modulation.** The new O-PCM also enables a broad class of tunable free-space optical devices capable of arbitrary phase or amplitude modulation for agile beam control in the infrared. As a proof-of-concept, we demonstrate reversible electrothermal switching of the GSS4T1 in a reflective pixel device on an 8″ CMOS compatible silicon wafer. Heat is applied to the material externally via joule heating of a Ti (50 nm)/Pt (20 nm) metal bilayer, which is separated from the GSS4T1 by a 10 nm $SiO_2$ film (Fig. 7a). The Pt coating prevents oxidation of the Ti electrode and contributes enhanced stability and cycling lifetime of the device. Pulse train profiles are applied to the pixel via the gate of a

power MOSFET connected in series to the device. Figure 7b shows the SEM image of a full device with the contact pads and Fig. 7c shows the zoom-in on the GSS4T1 patterned pixel. To monitor the state of the material, a 1550 nm laser was focused onto the pixel and the reflection was recorded on an InGaAs video camera with a frame rate of 100 frames per second. The time-dependent reflection measurement, shown in Fig. 7d, demonstrates a non-volatile absolute reflection change from 9 to 31%, which corresponds to over 240% relative change in reflection. The contrast can be further improved with an optimized multilayer film stack design. Our simulations predict that up to 30 dB contrast can be achieved in the short-wave infrared range. Raman measurement of the material structure confirms the electrically switched crystalline and amorphous peaks at 120 and 160 $cm^{-1}$, respectively, as shown in Fig. 7e. Over 1000 switching cycles have been demonstrated in these types of devices with consistent reflectance contrast, as discussed in Supplementary Note 8. Our electrical switching experimental setup does not allow time-resolved measurement of the phase transition process due to the camera's limited frame rate, and further study is warranted to fully characterize the crystallization rate of the GSST material.

## Discussion
In this article, we have demonstrated a new class of O-PCMs GSST engineered to achieve modulation based exclusively on the real part of the material refractive index and free from the loss penalty. The compositionally optimized alloy $Ge_2Sb_2Se_4Te_1$ claims an unprecedented material FOM over two orders of magnitude larger than that of classical GST alloys, benefiting from blue-shifted interband transitions as well as minimal FCA. A non-volatile optical switch was realized based on $Ge_2Sb_2Se_4Te_1$.

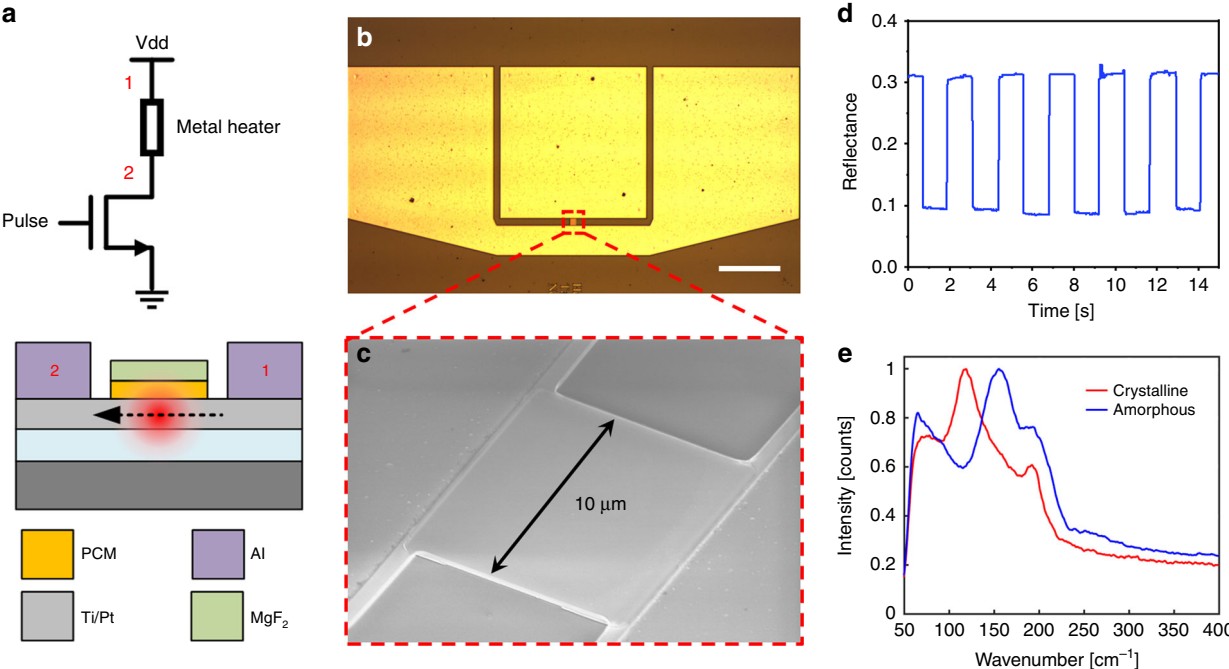

**Fig. 7** Electrothermal switching of Ge$_2$Sb$_2$Se$_4$Te$_1$ (GSS4T1). **a** Schematic of the device and test setup. **b** Top-view optical micrograph of the full device used to switch a 10 μm × 10 μm pixel. The three contact pads were used ground-source-ground electrical contacts. Scale bar: 100 μm. **c** Zoom-in on the pixel with a square pattern of GSS4T1. **d** Time-dependent absolute reflection measurements of a 1550 nm laser focused onto the pixel. **e** Raman measurements of the GSS4T1 film after an electrical amorphization and crystallization pulse profile

Its record low loss and switching contrast, derived from the exceptional FOM of the material, qualify the device as a useful building block for scalable photonic networks. An electrothermally switched free-space reflective pixel was also fabricated on an 8″ wafer in a CMOS compatible process and demonstrated with a microsecond amorphization switching time. These results enable a new path for electrical free-space infrared light control for applications in spatial light modulators, tunable reflective spectral filters, subwavelength reflective-phased arrays, beam steering, holography, and tunable metasurfaces. Moreover, isoelectronic substitution with light elements, as illustrated in our example of GSST, points to a generic route in the search of new O-PCMs optimized for low-loss photonic applications[53].

## Methods

**DFT modeling**. Standard ab initio calculations within the framework of density functional theory were performed using the Vienna Ab Initio Simulation Package (VASP v5.4)[54]. Plane-wave and projector-augmented-wave (PAW)[55] type pseudopotentials were applied with the electronic configurations of Ge: $4s^24p^2$, Sb: $5s^25p^3$, Te: $5s^25p^4$, Se: $4s^24p^4$, and a 300 eV kinetic-energy cutoff. Exchange-correlation effects were described with the GGA-PBEsol functional[56]. The structures were relaxed until all forces were smaller than 0.01 eV/Å. K-point grids of $12 \times 12 \times 4$, $12 \times 12 \times 1$, and $2 \times 9 \times 2$ were used for geometric optimization in hexagonal, cubic, and orthorhombic phases, while those for the electronic structure calculations are $24 \times 24 \times 8$, $24 \times 24 \times 2$, and $4 \times 18 \times 4$, respectively. The tetrahedron method with Blöchl corrections were employed to obtained total energies. While the cohesive energy strongly depends on the exchange-correlation functional employed in DFT, the relative energy differences between various phases and sequences are much less sensitive, enabling the comparison of stability between various configurations in this work—a method similarly adopted in previous studies[57]. Since the entropy contributions to the free energies have not been accounted in this work, and the standard DFT calculations only provide solutions at 0 K, our results cannot predict the driving force for transition between different structures accurately. Nevertheless, the variations of cohesive energy obtained with our approach are sufficient to qualitatively reveal the impact of Se substitution within each phase.

**Material synthesis**. Thin films of GSST were prepared using thermal evaporation from a single Ge$_2$Sb$_2$Se$_4$Te$_1$ source. Bulk starting material of Ge$_2$Sb$_2$Se$_4$Te$_1$ was

synthesized using a standard melt quench technique from high-purity (99.999%) raw elements[58]. The film deposition was performed using a custom-designed system (PVD Products, Inc.) following previously established protocols[59,60]. The substrate was not actively cooled, although the substrate temperature was measured by a thermocouple and maintained at below 40 °C throughout the deposition process. Stoichiometries of the films were confirmed using wavelength-dispersive spectroscopy (WDS) on a JEOL JXA-8200 SuperProbe Electron Probe Microanalyzer (EPMA) to be within 2% (atomic fraction) deviation from target compositions.

**Material characterizations**. Grazing incidence X-Ray diffraction (GIXRD) was performed using a Rigaku Smartlab Multipurpose Diffractometer (Cu Kα radiation) equipped with a high-flux 9 kW rotating anode X-ray source, parabolic graded multilayer optics and a scintillation detector. The GIXRD patterns were collected within one hour over a range of $2\theta = 10$–$80°$ at room temperature.

Before electrical conductivity and Hall measurement, Ti/Au (10/100 nm thickness) contacts were deposited using electron-beam evaporation through a shadow mask. A 3-nm-thick layer of alumina were then deposited on top of the GSST film as capping layer using atomic layer deposition to prevent film vaporization and surface oxidation. Hall and electrical conductivity measurements were carried out on a home-built Van der Pauw testing station. The samples were heat treated using a hotplate.

Optical properties of the films were measured with NIR ellipsometry from 500 to 2500 nm wavelengths in combination with a reflection and transmission Fourier transform infrared (FTIR) spectroscopy measurement from 2500 to 18,500 nm wavelengths. A gold mirror was used as the reference for the reflection measurement. Fitting of the real and imaginary parts of the refractive indices was performed with the Woollam WVASE software. The method allows us to quantify the material's imaginary part of refractive index down to below 0.01.

**In situ TEM analysis**. The sample was prepared on thin silicon nitride membranes with 2-μm holes, on which a 10 nm thick GSS4T1 film was deposited. Imaging was performed using Oxford's JEOL 2200 MCO aberration-corrected transmission electron microscope with CEOS (Corrected Electron Optical Systems GmbH) image corrector and an accelerating voltage of 80 kV. A heating holder (DENSsolutions) was used for in situ temperature control. All temperatures quoted in the manuscript regarding the TEM analysis are nominal values as given by the heating holder control, which can be slightly different from temperature of the sample on the SiN membrane due to thermal non-uniformity.

**Device fabrication**. The resonator devices and electrothermal switching devices were fabricated on silicon wafers with 3 μm thermal oxide from Silicon Quest

International. To fabricate the resonator devices, 400 nm SiN was first deposited using low pressure chemical vapor deposition. The resonators were patterned using electron-beam lithography on an Elionix ELS-F125 electron-beam lithography (EBL) system followed by reactive ion etching (CHF$_3$/CF$_4$ etching gas with 3:1 ratio at 30 mTorr total pressure). A 50-nm layer of GSS4T1 were then deposited and patterned using poly(methyl methacrylate) (PMMA) as the lift off resist and subsequently capped with 20 nm of SiO$_2$ deposited using plasma-enhanced chemical vapor deposition (PECVD). The electrothermal switching devices were fabricated from a 50-nm thick, evaporated tungsten film. Patterning of tungsten was achieved via reactive ion etching. 200-nm-thick aluminum contact pads were evaporated and patterned via lift off. 50 nm of GSS4T1 was then deposited, patterned via lift off, and encapsulated in 20 nm evaporated MgF$_2$.

**Integrated photonic device characterization**. The optical switch devices were measured on a home-built grating coupling system used in conjunction with an external cavity tunable laser (Luna Technologies) with a built-in optical vector analyzer. Laser light was coupled into and out of the devices using single-fiber probes. Details of the characterization setup can be found elsewhere[61]. The devices under test (DUT) were maintained at room temperature throughout the measurement.

**Laser-induced phase transition**. The laser system used to optically switch the phase change films consisted of a 633 nm and a 780 nm continuous-wave laser with a total optical power of 136 mW. An acoustic optical modulator with a 2 ns rise time was used to modulate the laser output to generate optical pulse trains. For amorphous to crystallization phase transitions, a pulse train with period of 1 μs, duty cycle of 0.03% (30 ns), and 100,000 repetitions was used. For crystalline to amorphous phase transitions, a single pulse with a width of 100 ns was used.

**Electrothermal switching**. In order to electrically amorphize GSS4T1, a single 1 μs pulse at 24 V is applied with a switching energy of 5.5 μJ. For crystallization, a pulse train consisting of 50 pulses with a period of 1 ms and duty cycle of 50% at 13 V is applied with a total switching energy of 42.5 mJ. Here the switching energy figures are quoted for 30 μm × 30 μm pixels, and we also experimentally demonstrated that the switching power can be reduced with smaller pixel sizes (Supplementary Note 9).

## Data availability

The data that support the findings of this study are available from the corresponding authors upon reasonable request.

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

## Acknowledgements

We would like to thank Jeffrey Grossman for helpful discussions and support to this work, and Skylar Deckoff-Jones for assistance with measurements. This material is based upon work supported by the Assistant Secretary of Defense for Research and Engineering under Air Force Contract No. FA8721-05-C-0002 and/or FA8702-15-D-0001, and by the Defense Advanced Research Projects Agency through the Defense Sciences Office (DSO) Program EXTREME Optics and Imaging (EXTREME) under Agreement No. HR00111720029 and the Young Faculty Award Program under Grant Number D18AP00070. We also acknowledge characterization facility support provided by the Materials Research Laboratory at MIT, as well as fabrication facility support by the Microsystems Technology Laboratories at MIT and Harvard University Center for Nanoscale Systems. J. Li acknowledges funding support from Shanghai Sailing Program (award No. 19YF 1435400). H. Li also acknowledges the support from the National Natural Science Foundation of China (11804403) and the Natural Science Foundation of Guangdong Province (2018B030306036). This research used computational resources of the National Super-computer Center in Guangzhou. Distribution statement: Approved for public release.

Distribution is unlimited. This material is based upon work supported by the Assistant Secretary of Defense for Research and Engineering under Air Force Contract No. FA8702-15-D-0001. Any opinions, findings, conclusions or recommendations expressed in this material are those of the author(s) and do not necessarily reflect the views of the Assistant Secretary of Defense for Research and Engineering. © 2018 Massachusetts Institute of Technology. Delivered to the U.S. Government with Unlimited Rights, as defined in DFARS Part 252.227-7013 or 7014 (Feb 2014). Notwithstanding any copyright notice, U.S. Government rights in this work are defined by DFARS 252.227-7013 or DFARS 252.227-7014 as detailed above. Use of this work other than as specifically authorized by the U.S. Government may violate any copyrights that exist in this work.

## Author contributions

Y.Z. deposited and characterized the material, modeled and fabricated the devices, and performed device measurement. J.B.C. and V.L. developed the laser and electrothermal switching method, performed thin film optical property characterizations, and helped with device design and fabrication. J.L. first developed the materials and device integration protocols. H. Li carried out the DFT modeling and interpreted the results. Q.D. conducted thin film structural analysis and assisted in device processing and testing. A.Y. and M.K. synthesized the bulk materials for thin film deposition. S.Z. and J.W. carried out the in situ TEM imaging and data analysis. Z.F., M.Y.S., C. Roberts, C. Ríos and P.R. contributed to device fabrication. H.Z., B.B., H. Lin and T.G. assisted in device measurement and data analysis. J.L. and J.H. conceived the project. T.G., J.W., V.L., K.R. and J.H. supervised the research. All authors contributed to technical discussions and writing the paper.

## Additional information

**Competing interests:** The authors declare no competing interests.

