## [Peer Review File · Nature Communications]

Editorial Note: Parts of this Peer Review File have been redacted as indicated to maintain the confidentiality of unpublished data and to remove third-party material where no permission to publish could be obtained.

Reviewers' comments:

Reviewer #2 (Remarks to the Author):

Review of revised manuscript for Nature Communications
“Extreme Broadband Transparent Optical Phase Change Materials for High-Performance Nonvolatile Photonics,” Yifei Zhang *et al.*

After comparing the revised manuscript with the comments of all three reviewers, it appears that the specific concerns of the reviewers have largely been met. The response from the authors now indicates an awareness that particularly for optical phase-changing materials (O-PCMs), figures of merit and other relatively straightforward criteria may not be appropriate for specific applications – because the nuances of a particular optical application may not be captured in one or two such general figures of merit. Yet even the present text sometimes seems to presume the contrary, as in the statement in the abstract that

“Current O-PCMs suffer from poor FOM’s on the order of unity, imposing a major hurdle towards **their** optical applications.” [Bold-face added.]

Yet there are O-PCMs – such as vanadium dioxide that the authors now, and properly, no longer compare with GSST – that incorporate small patches or nanostructures of thin O-PCM films into resonant structures (such as ring resonators) in order to achieve desirable functionalities.

The paper will be further strengthened if the authors will comb through the paper for statements that seem to imply universal applicability (or non-applicability) and change them to reflect these nuances that are so critical to photonic technologies. For example, the statement above could, and should, be changed to read

“Current O-PCMs suffer from poor FOM’s on the order of unity, imposing a major hurdle towards **their deployment in some** (or **specific**, or **many**) optical applications.” [Bold-face would be deleted.]

Reviewer #3 (Remarks to the Author):

Review of Zhang et al manuscript:

The authors have modified the paper by removing some of the claims from the previous version, added a bit more data on the optical properties of the GSST material. While this is not particularly striking, it shows some basic capability in terms of the optical performance. It still appears the authors cannot take full advantage of the large FOM due to several other limitations of their material or unknowns at the present due to their experimental capabilities. Also as the authors claim there are no established metrics for universal comparison, in which case I do feel the authors should compare and contrast their work to the original 'extreme broadband PCM materials' work by Yu and co-workers (Adv Mat, 28, 9117, 2016) as the application space does overlap in terms of metasurfaces (similar to what the authors have demonstrated here).

Another comment:

- The authors claim sub microsecond switching, but none of the data shows this. It will be useful for the authors to show this data, and how this is used in their devices. Both figures in main text file show data in the order of seconds, about 10^6 time scale larger than what they claim in the response letter.

- The authors claim a short pulse laser switched the material to another state suggesting the time scale of the laser pulse corresponds to the switching. But this does not necessarily make sense as the energy dumped by the laser can be dissipated over a much larger timescale. Unless I understand this incorrectly, the probe measurement must be conducted at the sub-microsecond or whatever time the authors claim using some other means. Is this what the authors have measured and I have missed in their manuscript? If so, this data should be shown in the paper.

These points continue to be the weak link in the paper in my opinion. There is no doubt tweaking the composition of GST materials (that have already been established as interesting candidates for O-PCMs) will result in some interesting property changes. Whether these tweaks result in some unexpected finding or a significant improvement over existing state-of-the-art is the question. I am still a bit confused about this part (given extensive previous works on Se doping of GST compounds) and it will be helpful for the authors to improve clarity on what they have measured in terms of the switching times (on-off and off-on) and how this is limited by the minimum thickness/volume needed for demonstrating optical property changes necessary for an interesting optical device.

Reviewer #4 (Remarks to the Author):

The present manuscript by Yifei Zhang and coworkers discusses a very timely topic, the application of optical phase change materials for photonics. They demonstrate that the identification of a novel compound obtained upon partially replacing Te in $\text{Ge}_2\text{Sb}_2\text{Te}_5$ by Se. This claim is supported by the corresponding figure of merit for merit, which is record-high for the present compound.

Furthermore, the manuscript is very well written and the authors have worked hard to address all questions raised in a previous round of reviews. Hence, I am in principle I favor of publication, since the present data should be available for a wider public. Nevertheless, in its present version, the manuscript still has some shortcomings and not all answers to the last round of reviews were fully satisfactory, from my more distant view.

The following points should thus be addressed:

- a) The authors argue on p. 2 of their manuscript, that the optical property modulation stems from a metal – insulator transition, which accompanies the structural transition. This statement is incorrect. There is indeed an MIT, and crystalline chalcogenides like $\text{Ge}_2\text{Sb}_2\text{Te}_5$ have indeed rather unconventional transport properties as correctly mentioned in the present manuscript. However, the change of optical properties is related to a change in bonding mechanism as first (to my knowledge) pointed out by Huang and Robertson [Phys. Rev. B 81, (2010)]. This claim has been further strengthened by recent work, which reveals that crystalline chalcogenides utilize a novel bonding mechanism [Advanced Material 30, 180377 (2017)]. In a follow up study, a map has been introduced, which helps localize potential compounds which utilize this bonding mechanism [Advanced Material 31, 1806280 (2018)]. These papers seem important here, since they provide a framework to discuss the present data and material system. Specifically, the following questions are raised:
 - 1) Does the material discussed here, i.e. $\text{Ge}_2\text{Sb}_2\text{Se}_4\text{Te}_1$, have the same property fingerprint as typical materials utilizing this bonding mechanism (high values of Z^* and ϵ^∞ , a soft lattice as evidenced by low values of the Grüneisen parameter, etc.). One could even wonder if the material shows the same unconventional bond breaking which has been reported for all crystalline compounds which employ this unconventional bonding mechanism [Advanced Materials 30, 1706735 (2018)].
 - 2) The material discussed here either shows such a drastic change of bonding, or not. No matter what the answer is, this would be important. For example, if the material would not show this significant change of bonding upon crystallization, this would immediately raise the question, where the pronounced optical contrast is coming from. This would hint towards a difference in density of both phases (amorphous vs. crystalline). However, this aspect is to my knowledge not even mentioned in the manuscript. A pronounced density change, if it exists, could also be indicative for issues with cyclability.
- b) One of the previous referees already raised the question of the switching speed. While the authors comment rightfully, that a more stable amorphous phase would relax the stringent requirements needed to amorphize upon laser exposure, a very stable amorphous phase would be unpleasantly slow to crystallize. Hence, the crystallization kinetics should either be measured, or it should be mentioned that this is a potential issue and needs to be studied in future work.
- c) There seems to be a problem with the DFT calculations. The authors argue that $\text{Ge}_2\text{Sb}_2\text{Se}_5$ has an orthorhombic ground state, a result that is in line with my expectations and the behavior of similar systems. However, in page 1.d, the results of the DFT calculations imply that the hexagonal and cubic phase are more stable. This seems rather inconsistent and confusing. Please clarify this point.
- d) The discussion of quantum confinement on p. 4 is surprising, too. The paper by Huang and Robertson, which focusses on the alignment of p-orbitals seems to provide a much simpler

and more convincing approach to explain the present data. If the authors disagree it would help if they could provide a quantitative measure of quantum confinement. The most convincing answer to this problem would come from a measurement of the optical matrix element. From the imaginary part of the dielectric function (Fermi's golden rule) the quality of the p-orbital alignment can be estimated.

- e) The authors conclude that they provide a new road for novel O-PCMs. It seems fair to say that the same can be said about [Advanced Material 31, 1806280 (2018)].

Response to Reviewer Comments NCOMMS-19-14118-T

Reviewer #2 (Remarks to the Author):

After comparing the revised manuscript with the comments of all three reviewers, it appears that the specific concerns of the reviewers have largely been met. The response from the authors now indicates an awareness that particularly for optical phase-changing materials (O-PCMs), figures of merit and other relatively straightforward criteria may not be appropriate for specific applications – because the nuances of a particular optical application may not be captured in one or two such general figures of merit. Yet even the present text sometimes seems to presume the contrary, as in the statement in the abstract that “Current O-PCMs suffer from poor FOM’s on the order of unity, imposing a major hurdle towards their optical applications.”

Yet there are O-PCMs – such as vanadium dioxide that the authors now, and properly, no longer compare with GSST – that incorporate small patches or nanostructures of thin O-PCM films into resonant structures (such as ring resonators) in order to achieve desirable functionalities.

The paper will be further strengthened if the authors will comb through the paper for statements that seem to imply universal applicability (or non-applicability) and change them to reflect these nuances that are so critical to photonic technologies. For example, the statement above could, and should, be changed to read “Current O-PCMs suffer from poor FOM’s on the order of unity, imposing a major hurdle towards their deployment in some (or specific, or many) optical applications.”

Response:

We thank the reviewer for the constructive comments. We have made several changes to wording in the manuscript following the reviewer’s suggestion (see red-lined version of the revised manuscript).

Reviewer #3 (Remarks to the Author):

[redacted]

[redacted]

We want to further point out that for integrated photonics applications, the loss penalty can be even more severe due to the need to propagate light through O-PCM films in the in-plane direction, and in many cases also through multiple cascaded devices. There are certainly applications where loss is not an issue or even essential (e.g. for analog optical computing architectures where optical loss is used to encode information³), but a zero-loss O-PCM enabling phase-only modulation is still the Holy Grail for a wide range of applications some of which we named in our previous response.

Another comment:

- The authors claim sub microsecond switching, but none of the data shows this. It will be useful for the authors to show this data, and how this is used in their devices. Both figures in main text file show data in the order of seconds, about 10^6 time scale larger than what they claim in the response letter.

Response:

To clarify, we had not claimed sub-microsecond switching in our previous response (see below). The reviewer also misinterpreted our data in the main text (Fig. 7d). In the figure, the reflectance trace consists of plateaus connected by vertical lines. The vertical lines correspond to when the switching takes place or when the bias is applied. Due to the limited frame rate of our camera (100 fps), we were unable to perform detailed time-resolved measurement of the actual switching process, and hence the transition appears as vertical lines in the plot. The steps correspond to the one-second intervals between switching operations (i.e. without voltage pulse applied) and does not correlate with the switching time. The fact that the steps are nearly flat confirms the

nonvolatile nature of the switching – i.e. the optical state is maintained without external power supply.

- The authors claim a short pulse laser switched the material to another state suggesting the time scale of the laser pulse corresponds to the switching. But this does not necessarily make sense as the energy dumped by the laser can be dissipated over a much larger timescale. Unless I understand this incorrectly, the probe measurement must be conducted at the sub-microsecond or whatever time the authors claim using some other means. Is this what the authors have measured and I have missed in their manuscript? If so, this data should be shown in the paper.

Response:

We want to clarify that “the time scale of the laser pulse corresponds to the switching” is not what we had stated or claimed. Quoting verbatim our previous response: “the minimal switching time is comparable or shorter than the thermal time constant, which is in the order of microseconds”, it is thus not possible to quantify the intrinsic switching speed from the single laser pulse switching experiment due to the thermal time constant limit. We have included below the time-dependent optical transmittance change at 1064 nm (probe) wavelength after a single laser pulse (pump). In this experiment, the blanket GSST film is deposited on an optically transparent CaF_2 substrate. This configuration, while suitable for in-situ transmission measurement, is not compatible with our electrical switch design due to the substrate choice. The trace (blue) can be fitted with a single exponential function (red) with a decay time of 2.4 μs . This decay time is consistent with our estimated thermal time constant of the system and therefore likely does not represent the intrinsic crystallization speed of the material.

Probe transmittance through a GSST film after a pump laser pulse at $t = 0$

These points continue to be the weak link in the paper in my opinion. There is no doubt tweaking the composition of GST materials (that have already been established as interesting candidates for O-PCMs) will result in some interesting property changes. Whether these tweaks result in some unexpected finding or a significant improvement over existing state-of-the-art is the

question. I am still a bit confused about this part (given extensive previous works on Se doping of GST compounds) and it will be helpful for the authors to improve clarity on what they have measured in terms of the switching times (on-off and off-on) and how this is limited by the minimum thickness/volume needed for demonstrating optical property changes necessary for an interesting optical device.

Response:

As we have elaborated in the previous response, the prior work only examines the alloys in the context of electronic memory or optical disc applications, never demonstrated the superior optical performance of Se-modified alloys, nor has a viable PCM composition for optical applications been identified. As we show in Fig. 5c, GSS4T1 is the only composition exhibiting large index modulation and broadband transparency with a FOM almost two orders of magnitude larger than other stoichiometries in the $\text{Ge}_2\text{Sb}_2\text{Se}_x\text{Te}_{5-x}$ system. This is a significant improvement over prior arts limited to only a few unoptimized compositions.

The critical thickness d_{\max} to allow complete re-amorphization of an O-PCM film is given by:

$$d_{\max} \sim \sqrt{\frac{\alpha \cdot \Delta T}{R_c}}$$

where α is the material thermal diffusivity, ΔT denotes the degree of supercooling corresponding to the fastest crystallization rate, and R_c the critical cooling rate of the material. According to reported critical cooling rate and thermal parameters of GST^{4,5}, we estimate that the critical thickness of GST alloys is around 60 – 100 nm. Uncoincidentally, the thickness of GST films in literature reports where reversible and complete switching has been demonstrated is maximally 70 nm⁶⁻¹⁰. This is much smaller than the wavelengths of light (in particular, near-IR telecom wavelengths and mid-IR wavelengths). The small thickness necessarily results in reduced optical confinement in O-PCM aka the active medium. For example, even if advanced Huygens resonator designs are implemented which enables remarkably thin meta-atoms with only one eighth of wavelength in thickness¹¹, meta-atoms operating in the infrared still cannot be made out of GST alone. Consequently, the spectral tuning range is drastically decreased (by approximately 70% at the telecom band, and even more in the mid-IR) due to the need to introduce other non-active materials.

Reviewer #4 (Remarks to the Author):

The present manuscript by Yifei Zhang and coworkers discusses a very timely topic, the application of optical phase change materials for photonics. They demonstrate that the identification of a novel compound obtained upon partially replacing Te in $\text{Ge}_2\text{Sb}_2\text{Te}_5$ by Se. This claim is supported by the corresponding figure of merit for merit, which is record-high for the present compound. Furthermore, the manuscript is very well written and the authors have worked hard to address all questions raised in a previous round of reviews. Hence, I am in principle I favor of publication, since the present data should be available for a wider public. Nevertheless, in its present version, the manuscript still has some shortcomings and not all answers to the last round of reviews were fully satisfactory, from my more distant view.

The following points should thus be addressed:

a) The authors argue on p. 2 of their manuscript, that the optical property modulation stems from a metal – insulator transition, which accompanies the structural transition. This statement is incorrect. There is indeed an MIT, and crystalline chalcogenides like $\text{Ge}_2\text{Sb}_2\text{Te}_5$ have indeed rather unconventional transport properties as correctly mentioned in the present manuscript.

However, the change of optical properties is related to a change in bonding mechanism as first (to my knowledge) pointed out by Huang and Robertson [Phys. Rev. B 81, (2010)]. This claim has been further strengthened by recent work, which reveals that crystalline chalcogenides utilize a novel bonding mechanism [Advanced Material 30, 180377 (2017)]. In a follow up study, a map has been introduced, which helps localize potential compounds which utilize this bonding mechanism [Advanced Material 31, 1806280 (2018)]. These papers seem important here, since they provide a framework to discuss the present data and material system. Specifically, the following questions are raised:

1) Does the material discussed here, i.e. Ge₂Sb₂Se₄Te₁, have the same property fingerprint as typical materials utilizing this bonding mechanism (high values of Z^* and ϵ_∞ , a soft lattice as evidenced by low values of the Grüneisen parameter, etc.). One could even wonder if the material shows the same unconventional bond breaking which has been reported for all crystalline compounds which employ this unconventional bonding mechanism [Advanced Materials 30, 1706735 (2018)].

2) The material discussed here either shows such a drastic change of bonding, or not. No matter what the answer is, this would be important. For example, if the material would not show this significant change of bonding upon crystallization, this would immediately raise the question, where the pronounced optical contrast is coming from. This would point towards a difference in density of both phases (amorphous vs. crystalline). However, this aspect is to my knowledge not even mentioned in the manuscript. A pronounced density change, if it exists, could also be indicative for issues with cyclability.

Response:

We thank the reviewer for raising this very important point. We have followed the reviewer's suggestion and re-worded the manuscript to avoid the confusion. When we alluded to metal-insulator transition, we were referring to the electrical conductivity change and sign reversal of temperature coefficient of conductivity (TCR) upon transition¹², which are results of free carrier injection. The transition inevitably introduces free carrier absorption, which is what we seek to tackle in our work. Most of the refractive index change, however, should not be attributed to the metal-insulator transition but rather resonant bonding or more precisely metavalent bonding as the reviewer correctly pointed out¹³. To confirm that the index modulation of GSS4T1 indeed stems from a bonding configuration change, we have computed the Born effective charges for the hexagonal phase of GSS4T1 in the table below, wherein the type 1, 2, 3, 4 are associated with the atom type Te, Se, Ge, Sb, respectively, and the 3×3 tensor is averaged over all atoms within the same type.

type	1	born charge	
	-3.52502	0.00088	0.00094
	0.00000	-3.52680	-0.00046
	0.00000	0.00000	-0.84369
type	2	born charge	
	-6.38549	-0.00008	0.00194
	0.00000	-6.38505	0.00065
	0.00000	0.00000	-4.11323
type	3	born charge	
	6.97926	-0.00018	-0.00178
	0.00000	6.97942	-0.00057
	0.00000	0.00000	4.64949
type	4	born charge	
	7.55424	-0.00011	-0.00258
	0.00000	7.55410	-0.00051
	0.00000	0.00000	3.99882

In the graph below, we use the weighted average absolute charge value over all atoms, which are calculated to be (6.46, 6.46, 3.84)e. Considering the pseudo 2-D nature of the crystal, we use the values associated with the in-plane direction, namely 6.46e for hexagonal GSS4T1. The large Born charges lie within the range of metavalent shown in Fig. 2b of Wuttig *et al.*¹³. On the other hand, the large optical dielectric constant observed in our experiments also support the scenario of metavalent bonding.

[redacted]

Further evidence on origin of the index change in GSS4T1 is furnished by thickness and geometric dimension changes upon transition. We experimentally measured ~ 5% thickness change in blanket GSS4T1 films upon crystallization. For discrete meta-atoms, we measured on average 2–3% linear dimension shrinkage. The dimension change is similar to those reported values of GST¹⁴, and can hardly account for the large index change we measured. Therefore, we conclude that bonding configuration change must be the dominant mechanism for index modulation in GSST.

b) One of the previous referees already raised the question of the switching speed. While the authors comment rightfully, that a more stable amorphous phase would relax the stringent requirements needed to amorphize upon laser exposure, a very stable amorphous phase would be unpleasantly slow to crystallize. Hence, the crystallization kinetics should either be measured, or it should be mentioned that this is a potential issue and needs to be studied in future work.

Response:

We agree with the assessment and have added a statement about limitations of our current setup in performing dynamic measurements while emphasizing the importance of future tests along this direction.

c) There seems to be a problem with the DFT calculations. The authors argue that $\text{Ge}_2\text{Sb}_2\text{Se}_5$ has an orthorhombic ground state, a result that is in line with my expectations and the behavior of similar systems. However, in page 1.d, the results of the DFT calculations imply that the hexagonal and cubic phase are more stable. This seems rather inconsistent and confusing. Please clarify this point.

Response:

We thank the reviewer for pointing this out. We agree that the lower stability of the orthorhombic phase of $\text{Ge}_2\text{Sb}_2\text{Se}_5$ compared to those of other phases predicted by DFT calculations is inconsistent with experimental observation. Such discrepancy may arise from two reasons: 1) the intrinsic imperfections of exchange-correlation functional to describe the dependence of energy on density gradient and to address the derivative discontinuity of Kohn-Sham potential make standard DFT approaches inefficient to predict the relative stability of various phases with distinct bonding characters and similar cohesive energies. Previous study shows a variation of more than 200 meV/atom regarding the relative energies of β -Sn referred to diamond phase calculated with different exchange-correlation functionals¹⁵; 2) the DFT simulations determine the structures and energies at 0 K, while lattice phonon vibrational analysis is demanded to predict finite-temperature phase stability¹⁶. These issues definitely are worth further explorations.

In fact, similar problem has been encountered in prior DFT simulations on GST, wherein the calculated total energy of metastable cubic phase is slightly lower than that of stable hexagonal phase¹⁷. Nevertheless, the successful prediction of atomic sequences within each phase reported in the same work¹⁷ suggests that the DFT method is adequate to reveal the change of stability with respect to compositional variation.

To avoid confusion, we have added the following paragraph to the Supporting Information Section I:

“We note that the lower stability of the orthorhombic phase of $\text{Ge}_2\text{Sb}_2\text{Se}_5$ compared to those of other phases predicted by DFT calculations is inconsistent with experimental observation. Such discrepancy may arise from two reasons: 1) the intrinsic imperfections of exchange-correlation functional to describe the dependence of energy on density gradient and to address the derivative discontinuity of Kohn-Sham potential make standard DFT approaches inefficient to predict the relative stability of various phases with distinct bonding characters and similar cohesive energies; 2) the DFT simulations determine the structures and energies at 0 K, while lattice phonon vibrational analysis is demanded to predict finite-temperature phase stability.”

d) The discussion of quantum confinement on p. 4 is surprising, too. The paper by Huang and Robertson, which focusses on the alignment of p-orbitals seems to provide a much simpler and more convincing approach to explain the present data. If the authors disagree it would help if they could provide a quantitative measure of quantum confinement. The most convincing answer to this problem would come from a measurement of the optical matrix element. From the imaginary part of the dielectric function (Fermi's golden rule) the quality of the p-orbital alignment can be estimated.

Response:

We appreciate the valuable suggestion from the reviewer. After perusing the paper published by Huang and Robertson¹⁸ as well as other related literature, we are convinced that the variation of *p*-orbital alignment and the consequentially variation of resonance bonding character are dominantly responsible for the change of bandgap among different phases. According to previous DFT calculations, the optical gaps of amorphous Ge₂Sb₂Te₅, Ge₂Sb₂Te₄Se, and Ge₂Sb₂Te₂Se₃ are 0.24 eV, 0.29 eV, and 0.53 eV larger than their crystalline counterparts, respectively¹⁹. Such variations of optical gap are comparable to our calculated 0.3 eV increase in bandgap of orthorhombic Ge₂Sb₂Se₅ compared to that of the hexagonal phase, indicating that *p*-orbital alignment rather than quantum confinement effect plays the critical role in tuning the bandgap.

The inaccurate statement regarding quantum confinement effect at page 4 of main text has been revised as follows: "This is expected from the lacking of half-filled degenerate orbitals due to *p*-orbital misalignment²⁶. The charge density distributions (Figs. 2f and 2g) further reveal that misalignment between *p*-orbitals due to the surface curvature of atomic blocks and large amount of interstitial sites in the orthorhombic Ge₂Sb₂Se₅ phase eliminates the resonance bonding mechanism."

e) The authors conclude that they provide a new road for novel O-PCMs. It seems fair to say that the same can be said about [Advanced Material 31, 1806280 (2018)].

Response:

We concur though the paper's emphasis is not on optical applications. We have referenced this paper in the revised manuscript.

References

1. Li, Z. *et al.* Correlated Perovskites as a New Platform for Super-Broadband-Tunable Photonics. *Adv. Mater.* **28**, 9117–9125 (2016).
2. Hail, C. U., Michel, A. U., Poulidakos, D. & Eghlidi, H. Optical Metasurfaces: Evolving from Passive to Adaptive. *Adv. Opt. Mater.* 1801786 (2019). doi:10.1002/adom.201801786
3. Ríos, C. *et al.* In-memory computing on a photonic platform. *Sci. Adv.* **5**, eaau5759 (2019).
4. Lyeo, H.-K. *et al.* Thermal conductivity of phase-change material Ge₂Sb₂Te₅. *Appl. Phys. Lett.* **89**, 151904 (2006).
5. Adam, J.-L. & Zhang, X. (Scientific researcher). *Chalcogenide glasses : preparation, properties and applications*.
6. Wang, Q. *et al.* Optically reconfigurable metasurfaces and photonic devices based on phase change materials. *Nat. Photonics* **10**, 60–65 (2016).
7. Gholipour, B., Zhang, J., MacDonald, K. F., Hewak, D. W. & Zheludev, N. I. An All-Optical, Non-volatile, Bidirectional, Phase-Change Meta-Switch. *Adv. Mater.* **25**, 3050–3054 (2013).
8. Michel, A.-K. U. *et al.* Reversible Optical Switching of Infrared Antenna Resonances with Ultrathin Phase-Change Layers Using Femtosecond Laser Pulses. *ACS Photonics* **1**, 833–839 (2014).
9. Lee, S.-Y. *et al.* Holographic image generation with a thin-film resonance caused by chalcogenide phase-change material. *Sci. Rep.* **7**, 41152 (2017).
10. Siegel, J., Schropp, A., Solis, J., Afonso, C. N. & Wuttig, M. Rewritable phase-change optical recording in Ge₂Sb₂Te₅ films induced by picosecond laser pulses. *Appl. Phys. Lett.* **84**, 2250–2252 (2004).
11. Zhang, L. *et al.* Ultra-thin high-efficiency mid-infrared transmissive Huygens meta-optics. *Nat. Commun.* **9**, 1481 (2018).
12. Siegrist, T. *et al.* Disorder-induced localization in crystalline phase-change materials. *Nat. Mater.* **10**, 202–208 (2011).
13. Wuttig, M., Deringer, V. L., Gonze, X., Bichara, C. & Raty, J.-Y. Incipient Metals: Functional Materials with a Unique Bonding Mechanism. *Adv. Mater.* **30**, 1803777 (2018).
14. Nazeer, H., Bhaskaran, H., Woldering, L. A. & Abelmann, L. Young's modulus and residual stress of GeSbTe phase-change thin films. *Thin Solid Films* **592**, 69–75 (2015).
15. Hennig, R. G. *et al.* Phase transformation in Si from semiconducting diamond to metallic β-Sn phase in QMC and DFT under hydrostatic and anisotropic stress. *Phys. Rev. B* **82**, 014101 (2010).
16. Løvvik, O. M., Opalka, S. M., Brinks, H. W. & Hauback, B. C. Crystal structure and thermodynamic stability of the lithium alanates LiAlH₄ and Li₃AlH₆. *Phys. Rev. B* **69**, 134117 (2004).
17. Sun, Z., Zhou, J. & Ahuja, R. Structure of Phase Change Materials for Data Storage. *Phys. Rev. Lett.* **96**, 055507 (2006).
18. Huang, B. & Robertson, J. Bonding origin of optical contrast in phase-change memory materials. *Phys. Rev. B* **81**, 081204 (2010).
19. Koch, C. *et al.* Enhanced temperature stability and exceptionally high electrical contrast of selenium substituted Ge₂Sb₂Te₅ phase change materials. *RSC Adv.* **7**, 17164–17172 (2017).

REVIEWERS' COMMENTS:

Reviewer #3 (Remarks to the Author):

The authors have addressed in large part the comments from the previous reviews. Given the broad scope of phase change materials in photonics and absence of singularly important metrics that are relevant for a number of different applications due to the differing role of loss in each of these device technologies, I still believe it will be imperative to properly place their work in context of prior studies on 'extreme broadband' or 'super broadband' phase change materials and therefore discuss the Adv Mat (28, 9117, 2016) and Adv Mat (31, 1806280, 2018) papers and how this present study extends the earlier literature.

It will be useful for the authors to include the thermal relaxation data shown in referee response letter into their manuscript files to illustrate their rationale for the timescale explanations.

It will be important for the authors to discuss in the manuscript whether they are able to take advantage of the full FOM in GSST41 system and how this compromises the dynamical behavior. This could serve as motivation for future studies and also present a balanced view point on this material, while showing some benefits for improved FOM, has a long way to go before practical use due to several limitations on the crystallization kinetic limitations.

Reviewer #4 (Remarks to the Author):

The authors have carefully addressed the comments made by the referees (especially the more detailed comments from reviewer #3 and #4). Their answers are rather convincing. While it is clear (as also stated by the authors) that studies of the switching kinetics are still an important next steps; this could be the content of a follow-up paper and is not required for the present manuscript.

Also the comparison with SmNiO₂ is helpful showing the respective strengths and weaknesses of both materials.

I also appreciate that the authors determined Born effective charges, hence further supporting the unusual bonding in their material.

Their measurements of the density/thickness change further support this claim, that a change of bonding is responsible for the change in optical properties upon crystallization.

There is only one point that surprises me: the authors argue that reference Advanced Materials 31, 1806280 (2019) has been included. I could not find this reference in the manuscript, even though the authors mentioned it as being included. This reference seems to be important since it provides a novel approach to identify new phase change materials in a systematic fashion.

Response to Reviewer Comments NCOMMS-19-14118-T

Reviewer #3 (Remarks to the Author):

The authors have addressed in large part the comments from the previous reviews. Given the broad scope of phase change materials in photonics and absence of singularly important metrics that are relevant for a number of different applications due to the differing role of loss in each of these device technologies, I still believe it will be imperative to properly place their work in context of prior studies on 'extreme broadband' or 'super broadband' phase change materials and therefore discuss the Adv Mat (28, 9117, 2016) and Adv Mat (31, 1806280, 2018) papers and how this present study extends the earlier literature.

It will be useful for the authors to include the thermal relaxation data shown in referee response letter into their manuscript files to illustrate their rationale for the timescale explanations.

It will be important for the authors to discuss in the manuscript whether they are able to take advantage of the full FOM in GSST41 system and how this compromises the dynamical behavior. This could serve as motivation for future studies and also present a balanced view point on this material, while showing some benefits for improved FOM, has a long way to go before practical use due to several limitations on the crystallization kinetic limitations.

Response:

We concur that metrics vary for different applications. However, as we have shown in the previous response, two-state transparency is an ideal feature for many device configurations. The paper Adv Mat (28, 9117, 2016) discussed a broadband tunable material, rather than a broadband transparent material. We concur that the Adv Mat (31, 1806280, 2018) paper is an important reference. However, the paper's emphasis is not on optical applications. We therefore believe our paper is distinctive enough from the papers mentioned above.

The thermal relaxation data, as we had explained earlier in our response letter, likely do not reflect the switching speed of the material and rather manifest the thermal time constant of the film. The thermal time constant is dependent on the film thickness, stack configuration, and substrate material and adds little value to elucidating the intrinsic O-PCM properties. We therefore feel that it is largely irrelevant and inappropriate to include the data in the manuscript text.

Reviewer #4 (Remarks to the Author):

The authors have carefully addressed the comments made by the referees (especially the more detailed comments from reviewer #3 and #4). Their answers are rather convincing. While it is clear (as also stated by the authors) that studies of the switching kinetics are still an important next steps; this could be the content of a follow-up paper and is not required for the present manuscript.

Also the comparison with SmNiO₂ is helpful showing the respective strengths and weaknesses of both materials.

I also appreciate that the authors determined Born effective charges, hence further supporting the unusual bonding in their material.

Their measurements of the density/thickness change further support this claim, that a change of bonding is responsible for the change in optical properties upon crystallization.

There is only one point that surprises me: the authors argue that reference Advanced Materials 31,

1806280 (2019) has been included. I could not find this reference in the manuscript, even though the authors mentioned it as being included. This reference seems to be important since it provides a novel approach to identify new phase change materials in a systematic fashion.

Response:

We thank the reviewer for the comments and we do sincerely apologize for our negligence. There was an error with our reference management software which mistakenly quoted another paper from the same group. We have now included this paper as reference 28 in the revised manuscript.